# Targeted Mutations Produce Divergent Characteristics in Pedigreed Sake Yeast Strains

**DOI:** 10.3390/microorganisms11051274

**Published:** 2023-05-12

**Authors:** Norapat Klinkaewboonwong, Shinsuke Ohnuki, Tomoya Chadani, Ikuhisa Nishida, Yuto Ushiyama, Saki Tomiyama, Atsuko Isogai, Tetsuya Goshima, Farzan Ghanegolmohammadi, Tomoyuki Nishi, Katsuhiko Kitamoto, Takeshi Akao, Dai Hirata, Yoshikazu Ohya

**Affiliations:** 1Department of Integrated Biosciences, Graduate School of Frontier Sciences, The University of Tokyo, Chiba 277-8562, Japan; 8124564286@edu.k.u-tokyo.ac.jp (N.K.); ohnuki@edu.k.u-tokyo.ac.jp (S.O.); tc.biology0048@gmail.com (T.C.); farzang@mit.edu (F.G.); 2Sakeology Center, Niigata University, 2-8050, Ikarashi, Niigata 950-2181, Japan; i-nishida@sake.nu.niigata-u.ac.jp; 3Sakeology Course, Graduate School of Science and Technology, Niigata University, 2-8050, Ikarashi, Niigata 950-2181, Japan; f22d057j@mail.cc.niigata-u.ac.jp (Y.U.); f22d060j@mail.cc.niigata-u.ac.jp (S.T.); 4National Research Institute of Brewing, Higashi-Hiroshima, Hiroshima 739-0046, Japan; isogai@nrib.go.jp (A.I.); t.goshima@nrib.go.jp (T.G.); akao_t@nrib.go.jp (T.A.); 5Program of Biotechnology, Graduate School of Integrated Sciences for Life, Hiroshima University, 1-3-1 Kagamiyama, Higashi-Hiroshima 739-8530, Japan; 6Department of Biological Engineering, Massachusetts Institute of Technology, Cambridge, MA 02139, USA; 7Sake Research Center, Asahi Sake Brewing Co., Ltd., Nagaoka, Niigata 949-5494, Japan; nishitomoyuki@asahi-shuzo.co.jp; 8Department of Pharmaceutical and Medical Business Sciences, Nihon Pharmaceutical University, Bunkyo-ku, Tokyo 113-0034, Japan; k-kitamoto@nichiyaku.ac.jp; 9Collaborative Research Institute for Innovative Microbiology, The University of Tokyo, Tokyo 113-8657, Japan

**Keywords:** *Saccharomyces cerevisiae*, genome editing, sake yeast, *FAS2*, CRISPR/Cas9

## Abstract

Modification of the genetic background and, in some cases, the introduction of targeted mutations can play a critical role in producing trait characteristics during the breeding of crops, livestock, and microorganisms. However, the question of how similar trait characteristics emerge when the same target mutation is introduced into different genetic backgrounds is unclear. In a previous study, we performed genome editing of *AWA1*, *CAR1*, *MDE1*, and *FAS2* on the standard sake yeast strain Kyokai No. 7 to breed a sake yeast with multiple excellent brewing characteristics. By introducing the same targeted mutations into other pedigreed sake yeast strains, such as Kyokai strains No. 6, No. 9, and No. 10, we were able to create sake yeasts with the same excellent brewing characteristics. However, we found that other components of sake made by the genome-edited yeast strains did not change in the exact same way. For example, amino acid and isobutanol contents differed among the strain backgrounds. We also showed that changes in yeast cell morphology induced by the targeted mutations also differed depending on the strain backgrounds. The number of commonly changed morphological parameters was limited. Thus, divergent characteristics were produced by the targeted mutations in pedigreed sake yeast strains, suggesting a breeding strategy to generate a variety of sake yeasts with excellent brewing characteristics.

## 1. Introduction

Sake is a traditional Japanese alcoholic beverage made from rice, rice koji, and water, which are first fermented and later filtered [1]. Koji mold converts starch derived from rice into glucose, and sake yeast converts glucose into ethanol in the brewing process. Therefore, sake yeast cooperates with koji mold to produce alcohol, creating a taste and aroma unique to sake [2].

Sake yeast is a subspecies of domesticated *Saccharomyces cerevisiae*, being composed of closely related strains and sharing common characteristics. A neighbor-joining tree revealed that it is different from beer yeasts and wine yeasts, forming a close clade with Asian brewer’s yeast [3]. Molecular dating analysis suggested that sake yeasts diverged from other yeasts ~4000 years ago. Genetic features such as aneuploidy and loss of heterozygosity are frequently observed in sake yeast. There are many genes specific to sake yeast such as *AWA1* [4] and *MAL73* [5]. However, a detailed examination of genetic sequences [3] and phenotypes [6,7] revealed the diversity of sake yeasts. There are more than 1000 single-nucleotide polymorphisms (SNPs) between representative pedigreed sake yeast isolates. Sake yeasts with different genetic backgrounds have different characteristics such as optimal growth temperature and cell morphology. Thus, it has become clear that sake yeast is a group with both commonalities and diversity.

The use of excellent pedigrees with different genetic backgrounds is an effective breeding strategy (Figure 1A). When high-quality sake is produced in a sake brewery, the sake yeast harbored in the brewery is isolated from its moromi and used as an excellent sake yeast isolate [8]. The currently available sake yeasts with different genetic backgrounds were introduced in this manner, including Kyokai No. 6 (K6), No. 7 (K7), No. 9 (K9), and No. 10 (K10). These pedigreed sake yeast isolates are valued in different breweries, because they each have their own distinct brewing characteristics [9]. In addition, sake yeast strains with excellent brewing characteristics have been further developed from these representative strains. By crossbreeding and inducing mutations, Kyokai No. 13 (K13) [10] and No. 1801 (K1801) [11] were generated, making it possible to produce sake with a strong ginjo aroma. Thus, the breeding of sake yeast strains has mostly been carried out using the representative excellent sake yeasts K6, K7, K9, and K10 as starting materials.

Another breeding strategy uses an evidence-based approach. *AWA1* encodes the mannoprotein present in the cell wall that is involved in the formation of foam during sake fermentation. As the foam occupies a large volume in the upper part of the brewing tank, the development of non-foam-forming sake yeast via the inactivation of Awa1 is advantageous for sake fermentation [4]. *CAR1* encodes the arginase involved in urea production. As the potential carcinogen ethyl carbamate is derived from urea, the loss of *CAR1* activity is suitable for making safer sake [12]. *MDE1* [13] and *FAS2* [14] act in the methionine salvage pathway and fatty acid synthesis, respectively, and are involved in the production of flavor in sake. Recent genome-editing technology has made it possible to breed sake yeast strains with these excellent brewing characteristics that did not previously exist [15]. By targeting the genes *AWA1*, *CAR1*, *MDE1*, and *FAS2* and using K7 as a starting material, a non-foam-forming strain that makes sake without producing carcinogens or an unpleasant odor, while producing a sweet ginjo aroma, was developed (Figure 1B). However, similar approaches were not attempted in sake yeast strains with other genetic backgrounds, and therefore, it is unclear whether the same excellent brewing characteristics could be obtained in different background strains.

This study was undertaken by combining the two previously mentioned approaches—namely, the use of pedigreed sake yeast strains with different genetic backgrounds and the evidence-based approach (Figure 1C). The purpose of this study was to investigate the effects of the targeted mutations in representative sake yeast pedigrees. To understand the generality and specificity of the effects of the targeted mutations, we conducted genome editing of K6, K9, and K10 and analyzed brewing characteristics and morphological phenotypes.

## 2. Materials and Methods

### 2.1. Strains and Media

The sake yeast strains developed in this study were all derived from the foaming isolates K6, K7, K9, and K10. The genome-edited strains constructed in this study are shown in Appendix A. Yeast strains were cultivated at 30 °C in yeast extract peptone dextrose medium containing 1% (*w*/*v*) Bacto yeast extract (BD Biosciences, Palo Alto, CA, USA), 2% (*w*/*v*) Bacto peptone (BD Biosciences), and 2% glucose to transform yeast strains, extract yeast DNA, and prepare precultures for fermentation tests. Geneticin (Takara, Kyoto, Japan) was added at 350 μg/mL to yeast extract peptone dextrose agar plates containing 2% agar (Shouei, Tokyo, Japan) after autoclaving as described previously [15]. We applied genome-editing techniques to the *AWA1*, *CAR1*, *MDE1*, and *FAS2* genes in the K6, K7, K9, and K10 genomes as described previously [15]. We used the pCAS-Pro-URA3 plasmid-co-expressing nuclease protein Cas9 and Ribozyme-sgRNA, which guides the Cas9 to target sequences [16]. The pCAS-Pro-AWA1, pCAS-Pro-CAR1, pCAS-Pro-MDE1, and pCAS-Pro-FAS2 (G1250S) plasmids are available upon request.

### 2.2. Whole-Genome Sequencing

DNA was extracted from strains K6, K7, K9, K10, K6GE41, K7GE41, K9GE41, and K10GE41 to determine their whole-genome sequences. First, high-molecular-weight DNA (25–90 µg; ~24-kb fragments) was isolated with a Genomic-tip 100/G kit (Qiagen, Germantown, MD, USA) in accordance with the manufacturer’s instructions. The purity of the DNA samples was estimated using a spectrophotometer (NanoDrop; Thermo Fisher Scientific, Waltham, MA, USA) and electrophoresis. Whole-genome sequencing was outsourced to GeneBay, Inc. (Yokohama, Japan). The DNA samples were then sent to Novogene (Singapore) for the preparation of a PCR-free, paired-end sequencing library and whole-genome sequence analysis (2 × 150 bp) using an Illumina NovaSeq 6000 sequencing platform (Illumina, San Diego, CA, USA) at ~100-fold nominal coverage. The adapter contamination was removed, and the low-quality bases trimmed. The K7 reference genome (NRIB_SYGD, txid721032) was obtained from the Sake Yeast Genome Database (version 1.0; https://nribf1.nrib.go.jp/SYGD/, accessed on 22 August 2022) and prepared for use in sequencing data analyses. The software packages used for sequencing data analysis were: Sequence Alignment/Map Tools (version 1.0.8) [17] to convert the “sam” format to the “bam” format and to modify information concerning the paired reads; Burrows–Wheeler Aligner (version 0.7.17) [18] for mapping reads to the K7 reference genome; Picard-tools (version 2.18.29; http://broadinstitute.github.io/picard; accessed on 22 August 2022) to remove duplicate reads; Genome Analysis TK (version 4.2.6.1) [19] to rearrange the bam format, extract mutation candidates, identify and filter variants relative to K7, and identify the mutations; and finally, variants were annotated manually with snpEff software (version 5.1; https://pcingola.github.io/SnpEff/, accessed on 22 August 2022) [20]. All SNP data were then extracted into the CSV format using R software (https://cran.r-project.org/; accessed on 22 August 2022). First, SNPs in K6GE41, K9GE41, and K10GE41 that were duplicates of K6, K9, and K10 SNPs, respectively, were excluded. The remaining SNPs were then extracted based on their annotation impact and QC score. SNPs that are silent mutations (low annotation impact) or located in the non-coding region (modifier) were excluded in our final SNPs. Each extracted SNP was finally checked on the Integrative Genomics Viewers [21].

### 2.3. Analysis of Structural Changes Predicted Due to Mutations

Protein Data Bank (PDB) files of wild-type sequences of Pfa5, Mef1, and Xrn1 were downloaded from AlphaFold [22]. AlphaFold2 was then used to make PDB files of Flo5, Nap1, and Num1 (protein sequences were obtained from the Comprehensive Sake Yeast Genome Database; *Saccharomyces cerevisiae* Kyokai No. 7) as well as mutants using MMseqs2 [23] for multiple sequence alignments. The model with the highest per-residue accuracy of the structure was finally chosen. STRIDE [24] was used to extract secondary structures from the PDB files.

### 2.4. Component Analysis of Sake Made in Small-Scale Fermentation Tests

A sake mash was prepared by mixing 72.8 g of pregelatinized rice (corresponding to 100 g of white rice), 19.2 g of dried koji (rice with *Aspergillus oryzae* mold, corresponding to 20 g of white rice), 136 μL of 90% lactic acid, and 170 mL of water containing 1 × 10^9^ precultured yeast cells. The mash (three replicates) was incubated at 15 °C for 20 days without shaking. The fermentation was monitored every day by quantifying the amount of evolved CO_2_ by measuring the weight loss of the sake mash. After completion of the sake fermentation, the mash was collected in 50 mL centrifuge tubes and centrifuged at 15 °C and 5000 rpm for 15 min. The supernatant was filtered with microfiber cloth to yield sake product and stored at −80 °C.

Measurements of the sake meter value (SMV), ethanol concentration, ethyl acetate, 1-propanol, isobutanol (iBuOH), isoamyl alcohol, isoamyl acetate, ethyl caproate, acidity, and amino acid contents were obtained as described previously [25]. Briefly, SMV was calculated after measuring the density of the sake relative to water. The ethanol concentration was measured by gas chromatography. Aroma components, such as ethyl acetate, 1-propanol, iBuOH, isoamyl alcohol, isoamyl acetate, and ethyl caproate, were measured by gas chromatography. Acidity and amino acid contents were measured with an automatic titrator. Organic acids, including malic acid, succinic acid, lactic acid, citric acid, acetic acid, and phosphoric acid, were measured by liquid chromatography as described previously [15]. The precursor of dimethyl trisulfide (1,2-dihydroxy-5-(methylsulfinyl) pentan-3-one; DMTS-P1) was analyzed by liquid chromatography–mass spectrometry using (ethyl-d3)-DMTS-P1 as an internal standard, as described previously [15]. Urea was measured according to the method of [26]. Significant changes in the components were marked with asterisks for *p* < 0.01 (**) or *p* < 0.05 (*) after Tukey’s multiple comparison test.

Principal component analysis (PCA) with 21 parameters was performed on the mean values yielded from 24 samples (8 strains × 3 samples) of sake produced in the small-scale fermentation test (Appendix A). The mean value of each strain and the value of each sample were plotted with two components (cumulative contribution ratio 57.49%)

### 2.5. Fluorescence Staining, Microscopy, and Image Processing

Fluorescence staining procedures were as described previously [27]. Briefly, cells of yeast strains were cultivated until an early log phase (<5 × 10^6^) and fixed with medium containing 3.7% (*w*/*v*) formaldehyde (Wako, Osaka, Japan). We then triple-stained cells with fluorescein isothiocyanate-conjugated concanavalin A (Sigma, St. Louis, MO, USA) for the cell wall, rhodamine-phalloidin (Invitrogen, Carlsbad, CA, USA) for the actin cytoskeleton, and 4′,6-diamidino-2-phenylindole (Sigma) for nuclear DNA, as described previously [27]. Fluorescence microscopy images of the cells were acquired using a microscope (Axio Imager) equipped with a special lens (6100 Ecplan-Neofluar; Carl Zeiss, Oberkochen, Germany), a cooled-charge-coupled device camera (CoolSNAP HQ; Roper Scientific Photometrics, Tucson, AZ, USA), and appropriate software (AxioVision; Carl Zeiss). Micrographs of the cells were analyzed with image-processing software designed for diploid cells (CalMorph, version 1.3) [28] and further investigated statistically as described in the following section. Descriptions of each trait have been presented previously [27]. The CalMorph user manual is available at the *S. cerevisiae* Morphological Database (http://www.yeast.ib.k.u-tokyo.ac.jp/CalMorph/, accessed on 22 August 2022).

### 2.6. Morphological Phenotyping of Sake Yeasts

Morphological data obtained in this study (strains K6, K6GE01, K6GE21, K6GE31, K6GE41, K7, K7GE01, K7GE21, K7GE31, K7GE41, K9, K9GE01, K9GE21, K9GE31, K9GE41, K10, K10GE01, K10GE21, K10GE31, and K10GE41) were used for statistical analyses. The number of samples for K7, K7 derivatives, and the others were 37, 6, and 5, respectively. All statistical analysis was conducted using R software (https://cran.r-project.org/, accessed on 22 August 2022).

#### 2.6.1. Calculation of the Euclidean Distance in the Degenerated Morphological Space

The Euclidean distance [29] was used to assess morphological differences between two strains in the degenerated morphological space; this distance is near zero if the cell morphology of the two strains is similar but otherwise large. The 501-dimensional morphological data were degenerated into orthogonal phenotypic space by PCA. PCA was applied with a variance–covariance matrix to the Z-values of each sample in all strains for all 501 traits that were calculated by the Wald test for one-way analysis of variance (ANOVA) of a generalized linear model (GLM) with K7 as an intercept. From the PCA of all samples, the cumulative contribution ratios of the first 44 principal components (PCs) reached 90% (Appendix A). The Euclidean distance between each strain was calculated from the scores of the first 44 PCs (cumulative contribution ratio 90%), as described previously [7]. The mean values of K7 replicates (*n* = 37) were subtracted from all scores of the 44 PCs, then the square root of the sum of squares for the 44 PCs was calculated as the Euclidean distance. Finally, the difference in the Euclidean distance between the parental strain and the derivative (ex. K7GE31 and K7GE41, respectively) was assessed at FDR < 0.05 by the Wald test of the ANOVA model assuming a gamma distribution with a design matrix (Appendix A).

#### 2.6.2. Principal Component Analysis of the Effects of Genome Editing on Cell Morphology

Morphological changes in genome-edited strains were examined in different sake yeast pedigrees. After description using 501 parameters, they were normalized with a GLM using a probability distribution model [30]. An ANOVA model was applied using a design matrix (Appendix A) to detect genome-editing steps exhibiting significantly different morphological changes. Among the 501 morphological parameters, 452 were significantly different (likelihood ratio test, FDR = 0.05), and at least one genome-editing step was detected in 325 parameters (Wald test, FDR = 0.05). By taking account of the direction of the morphological change parameters, a Venn diagram was drawn to show the common features among the sake yeast pedigrees, where some parameters were counted twice when the direction of the change was reversed among the sake yeast pedigrees. For further analyses, replicates of all strains were converted into the *Z*-values of the Wald test for each parameter using the mean of the parent strain as a reference point.

To investigate the distribution of genome-edited strains, PCA was performed using the variance–covariance matrix for the mean values of each strain, and the first two components explained 65% of the variance (Appendix A). By mapping the *Z*-value of each replicate of all strains to this orthogonal space, we obtained the morphological distribution of the genome-edited strains.

Morphological phenotypes of genome-edited strains were characterized after performing PCA with a variance–covariance matrix. The first eight PCs covered 90% of the variance (Appendix A). When one-way ANOVA was applied to the 16 genome-edited strains with K7 as a reference, the model was significant for all eight PCs (likelihood ratio test, *p* < 0.01 after Bonferroni correction), implying significant morphological changes. Significantly changed traits of each strain were detected by Wald test with FDR < 0.01 and are shown in Appendix A. The morphological parameters correlated with each PC were identified by two-step successive PCA [31] and recognized as representative morphological features. The features of the eight PCs in the first-step PCA were represented by significantly correlated morphological parameters (FDR = 0.05, Appendix A). Then, for the parameters significantly correlated with each PC, a second PCA was performed using the *Z*-values of the 37 replicates of K7, yielding PCs that explain 60% of the variance, as shown In Appendix A. By this second-step PCA, representative morphological features of the eight PCs were extracted as morphological parameters with significant PC loadings (*p* < 0.01 after Bonferroni correction by uncorrelation test, Appendix A). Finally, the morphology correlated to each first-step PC is summarized in Appendix A.

## 3. Results

### 3.1. Isolation of Genome-Edited Sake Yeast Strains in K6, K9, and K10

We previously performed four-step serial breeding using genome-editing technology to create a sake yeast strain with excellent brewing characteristics. We employed a standard sake yeast isolate, K7, as a starting material [15,32] and constructed a final genome-edited strain, K7GE41, harboring eight mutations including *awa1*∆*/awa1*∆, *car1*∆*/car1*∆, *mde1*∆*/mde1*∆, and *FAS2* (G1250S)/*FAS2* (G1250S). In this study, using the same genome-editing strategy but different pedigreed strains—K6, K9, and K10—we constructed sake yeast strains K6GE41, K9GE41, and K10GE41, respectively (Appendix A). Whole-genome DNA sequencing of K7GE41 indicated that one heterozygous missense mutation and one loss of heterozygosity (LOH) were unexpectedly introduced into the K7 genome during the four-step breeding process (Appendix A). These genetic perturbations were predicted to have little or no effect on the brewing and morphological characteristics [15]. This study also revealed a small number of unexpected mutations in the genome-edited diploid strains: we found three, four, and three genes mutated only heterozygously in the K6GE41, K9GE41, and K10GE41 genomes, respectively (Appendix A). There was no common characteristic among the mutations introduced, suggesting that unexpected mutations were not due to off-target modifications by CRISPR/Cas9. Among the genes with unexpected mutations, four genes—*ZEO1*, *CHD1*, *EFT1*, and *SGE1*—are known to have no effect on fitness or morphology in laboratory strains (Appendix A) [27,33]. Although a deletion of the remaining six genes in the haploid affects growth and/or morphology, structural modeling analysis revealed that only the missense mutation in Nap1 affected the 3D structure of the protein (Appendix A). However, heterozygous deletion of *NAP1* resulted in no obvious effects on fitness or morphology [34,35]. These analyses suggested that the few unexpected mutations introduced during the genome-editing processes do not significantly affect the phenotypes of sake yeast strains.

### 3.2. Excellent Brewing Characteristics of the Genome-Edited Strains

We next analyzed the components of sake made by the genome-edited strains after performing small fermentation tests. The targeted *car1*∆/*car1*∆ mutations cause a defect in urea production. Sake made by K6GE41, K7GE41, K9GE41, and K10GE41 with the *car1*∆/*car1*∆ mutations all contained decreased amounts of urea (Figure 2A). Since the potential carcinogen ethyl carbamate is converted from urea during the storage of sake, the genome-edited yeasts with the *car1*∆/*car1*∆ mutations produce sake with improved safety, containing less ethyl carbamate [12]. The targeted *mde1*∆/*mde1*∆ mutations cause a defect in the production of DMTS, a major unpleasant component of hineka, and its precursor DMTS-P1 [13]. As a result, sake made by the genome-edited yeasts with the *mde1*∆*/mde1*∆ mutations were found to contain a decreased amount of DMTS-P1 (Figure 2B). Thus, all genome-edited yeast stains generated in this study exhibited decreased production of DMTS. Finally, the targeted *FAS2* (G1250S)/*FAS2* (G1250S) mutations enhance the production of ethyl caproate, producing sake with a strong ginjo aroma [14]. Sake made by the genome-edited yeasts with the *FAS2* (G1250S)/*FAS2* (G1250S) mutations all contained increased amounts of ethyl caproate (Figure 2C). Therefore, these results indicated that we succeeded in generating genome-edited strains in different genetic backgrounds to make sake without producing carcinogens or an unpleasant odor, while producing a sweet, ginjo aroma.

### 3.3. Component Analysis of Sake Made by the Genome-Edited Strains

Our previous studies indicated that introduction of the *FAS2*(G1250S)/*FAS2*(G1250S) mutations in K7GE41 significantly alters a total of 16 components in sake, including expected changes such as an increase in ethyl caproate and unexpected changes such as a decrease in SMV and 1-propanol (nPrOH) [15]. We still observed similar changes in 12 of 16 components with K7GE41, although we used a different lot of sake mash in this study, partially confirming experimental reproducibility (Figure 2, Figure 3 and Appendix A). We found that the component changes in sake made with K7GE41 were not always seen in sake made with other genome-edited strains. For example, although SMV and nPrOH were significantly reduced in all genome-edited strains (Tukey’s multiple comparison test, *p* < 0.05, Figure 3A,B), the amino acid content was significantly increased only in sake made with K7GE41 and K10GE41 (*p* < 0.05, Figure 3C). Isobutyl alcohol (iBuOH) was significantly decreased only in sake made by K6GE41, K7GE41, and K10GE41 (*p* < 0.01, Figure 4D). Changes in citric acid and phosphoric acid were also strain dependent (Appendix A). These results suggested that even if the same mutations are introduced, the composition of sake made by the strains other than K7GE41 changes differently.

To visually characterize the overall changes in sake components, we examined sakes made by all genome-edited strains (K6GE41, K7GE41, K9GE41, and K10GE41) and original strains (K6, K7, K9, and K10) in the degenerated orthogonal space after performing PCA. When we performed PCA with the 21 components, the contributions of PC1 and PC2 accounted for 36.58% and 20.91% of the variance, respectively (Appendix A), and the cumulative contribution was 57.49% (Appendix A). Examination of the changes from the original strains to the corresponding genome-edited strains revealed that all were generally increased on PC1 but with somewhat different directions (arrows in Appendix A). Urea, DMTS-P1, and ethyl caproate as well as SMV and nPrOH had higher absolute PC1 loadings values (Appendix A), implying that PC1 reflects commonly changing components. By contrast, the absolute values of PC2 and PC3 loadings were high in iBuOH (0.558) and amino acid content (0.417), respectively, which differed among the strain backgrounds (Figure 3C,D), meaning that differences among the strain backgrounds appear in other components than PC1 (Appendix A). These results suggested that even if the same mutations are introduced, the changes in the components of sake differ according to the strain background.

### 3.4. Morphological Analysis of Genome-Edited Sake Yeast Strains

We previously analyzed the morphological phenotypes of the genome-edited sake yeast strains generated in the K7 background. We found that *FAS2*(G1250S)/*FAS2*(G1250S) in K7GE41 causes the greatest morphological changes, including larger cell size [15]. After triple-staining the cell wall, actin, and nuclear DNA, we first examined the morphologies of genome-edited strains and the original strains and found that K7GE41 became large (Figure 4), as previously reported. The morphological phenotypes of the genome-edited strains were similar in other independent isolates (Appendix A), confirming the reproducibility of the morphological phenotypes. We then conducted high-dimensional morphological phenotyping of 16 successive genome-edited strains as well as their original strains using the image analysis system CalMorph as shown below. Holistic morphological abnormality (HMA) is a measure of how large a morphological change has occurred [34]. Examination of HMA at each step in different genetic backgrounds revealed that the largest morphological change occurred when K7GE41 was generated (*p* < 0.01 after the Bonferroni correction by likelihood ratio test of one-way ANOVA with gamma distribution, Figure 5A and Appendix A). Significant morphological changes were also observed when K6GE41 and K10GE41 were generated (*p* < 0.05, Figure 5A). At the time of generation of K9GE41, however, there was no detectable change in HMA. We next examined the positional relationships between the genome-edited strains in the degenerated orthogonal morphological space after performing PCA. As expected from the HMA analysis, K7GE41 was the furthest from the original K7 strain (Figure 5B). We also found that directions from the origin varied among the final genome-edited strains (K6GE41, K7GE41, K9GE41, and K10GE41), suggesting that the morphological changes due to *FAS2*(G1250S)/*FAS2*(G1250S) were different in the strain backgrounds.

We next examined the high-dimensional morphological changes at each step of breeding. For this purpose, first, the likelihood ratio test with the design matrix (Appendix A) and Wald test were used to detect the parameters for which each step of breeding had a significant effect (FDR = 0.05). Among the 501 morphological parameters, 325 parameters were detected with at least one step (Appendix A). A Venn diagram (Figure 6 and Appendix A) shows how many common morphological traits were changed in the same step of breeding in different backgrounds, clearly indicating that different morphological changes were observed depending on the backgrounds. In particular, when *FAS2*(G1250S)/*FAS2*(G1250S) was introduced, only 18 traits changed in common among the four background strains, whereas 94 traits changed only in the K7 background (Figure 6). The morphological changes in each step were summarized by PCA on Z-values of the parameters standardized with the Wald test. Schematic illustration of the morphological features of the final genome-edited strains (K6GE41, K7GE41, K9GE41, and K10GE41) showed that the morphology differs greatly due to the difference in the strain backgrounds (Figure 6).

## 4. Discussion

We investigated the effects of targeted mutations on different genetic backgrounds of sake yeast strains. The excellent brewing characteristics were reproduced in each pedigreed sake yeast strain, confirming that the evidence-based approach worked well for the breeding of sake yeast. By contrast, differences in other brewing characteristics and morphological phenotypes were observed when the same mutations were introduced into the different backgrounds, indicating it is not always possible to observe a one-to-one relationship between target mutations and phenotypes. We instead observed divergent characteristics due to the targeted mutations in different backgrounds. Taken together, we should breed with various sake yeast strains to achieve diverse brewing characteristics. This study provides an important contribution to the discussion on breeding strategies for domesticated microorganisms.

Since sake yeast strains with the targeted mutations were generated by genome-editing technology, we easily introduced the desired homozygous mutations into diploid sake yeast strains. A previous study indicated that all the 400 genome-edited strains contained desired homozygous mutations [15]. The genome-edited strains generated in this study had no unexpected homozygous mutations. We believe it unlikely that a few heterozygous mutations unexpectedly introduced during genome editing resulted in brewing characteristics or morphological changes. It should be noted that almost the same morphological changes were observed in the independently generated genome-edited strains, implying the reproducibility of morphological phenotypes due to the targeted mutations. Taken together, we concluded that the diverse characteristics were caused by the introduction of the same targeted mutations into different genetic backgrounds.

Previous studies with laboratory yeast strains revealed that the phenotypic consequence of a given mutation can be influenced by the genetic background [36,37]. Conditional gene essentiality occurs when the loss of function of a gene causes lethality in one genetic background but not another. However, the frequency of differences in gene essentiality found between laboratory strains S288c and Σ1278b was low, with only about 1% of genes being conditionally essential [36]. Given that there are 25,298 SNPs between these two strains [38], the genetic diversity of pedigreed sake yeast strains is 10 times lower than between S288c and Σ1278b. Brewing characteristics and morphological phenotypes may be more susceptible to genetic background than gene essentiality.

Commercial yeast such as sake yeast has a low sporulation rate, and breeding by crossing has long been considered difficult. Recently, a method called return-to-growth (RTG) was developed to generate many recombinants by returning to a nutrient medium during sporulation in medium containing potassium acetate [39,40]. RTG would generate sake yeast strains more diverse than the pedigreed sake yeast strains. In this study, breeding was performed by applying an evidence-based approach in different genetic backgrounds. In the future, it may be possible to combine an evidence-based approach with RTG to breed a variety of sake yeasts with more excellent brewing characteristics.

## 5. Conclusions

*AWA1*, *CAR1*, *MDE1*, and *FAS2* were introduced by genome-editing technology into pedigreed sake yeast strains with different genetic backgrounds, including K6, K7, K9, and K10. We succeeded in generating genome-edited yeast strains with excellent brewing characteristics in each pedigreed sake yeast strain, confirming the utility of the evidence-based approach. However, we also observed divergent brewing characteristics and morphology due to the targeted mutations in different backgrounds of sake yeast. Our results provide important insights for discussing the effects of breeding on domesticated microorganisms.

## Figures and Tables

**Figure 1 microorganisms-11-01274-f001:**
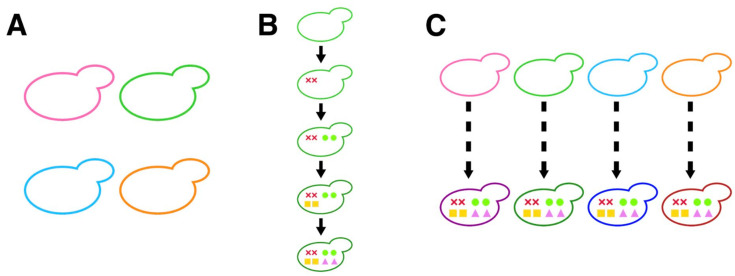
Breeding strategy of sake yeast. (**A**) Use of excellent pedigrees with different genetic backgrounds. Yeasts colored with hot pink (K6), lime green (K7), deep sky blue (K9), and dark orange (K10) represent different background yeast strains. (**B**) Introduction of the targeted mutations for breeding. An example showing the introduction of eight mutations in a sake yeast strain with the evidence-based approach. Colored marks (cross marks (*AWA1*), circles (*CAR1*), squares (*MDE1*), and triangles (*FAS2*(G1250S))) represent targeted mutations. (**C**) Combinatory use of pedigreed sake yeast strains and the evidence-based approach. Examples show the introduction of eight mutations in different genetic background strains.

**Figure 2 microorganisms-11-01274-f002:**
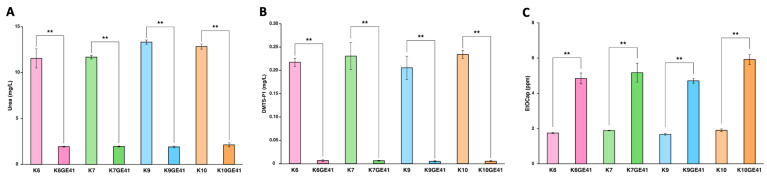
Measurement of urea (**A**), dimethyl trisulfide-P1 (DMTS-P1) (**B**), and ethyl caproate (**C**) in sake made with genome-edited strains. Error bars indicate standard error (*n* = 3). Asterisks indicate Tukey’s multiple comparison test (** *p* < 0.01).

**Figure 3 microorganisms-11-01274-f003:**
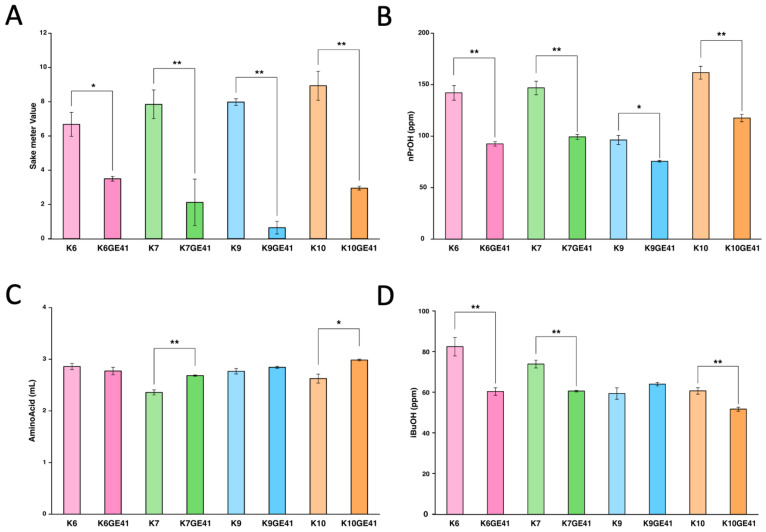
Measurement of sake meter value (**A**), 1-propanol (nPrOH) (**B**), amino acid (**C**), and isobutyl alcohol (iBuOH) (**D**) in sake made with genome-edited strains. Error bars indicate standard error (*n* = 3). Asterisks indicate Tukey’s multiple comparison test (* *p* < 0.05, ** *p* < 0.01).

**Figure 4 microorganisms-11-01274-f004:**
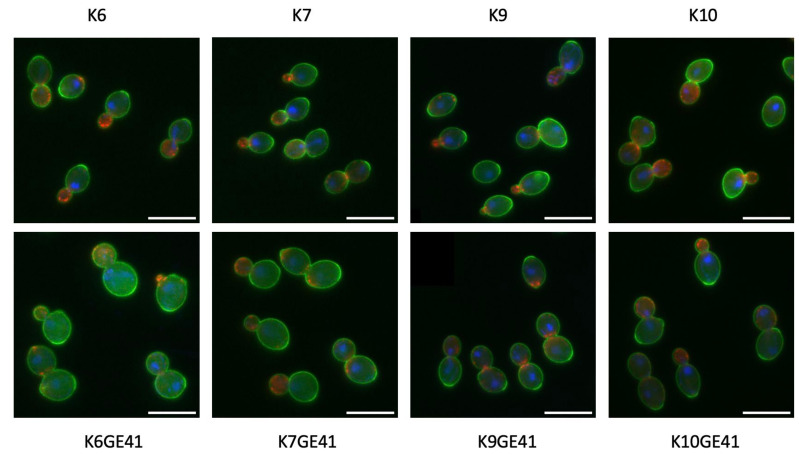
Morphological phenotypes of genome-edited yeast strains (K6GE41, K7GE01, K9GE41, and K10GE41) and their original pedigreed strains (K6, K7, K9, and K10). The cell wall (green), nuclear DNA (blue), and actin cytoskeleton (red) were stained with FITC-ConA, DAPI, and rhodamine-phalloidin, respectively, and observed under a microscope equipped with a special lens, a cooled charge-coupled device camera, and AxioVision software. Three photographs of the same field of view (green, cell wall; red, actin; and nuclei, blue) were superimposed on each other. Scale bars indicate 10 μm.

**Figure 5 microorganisms-11-01274-f005:**
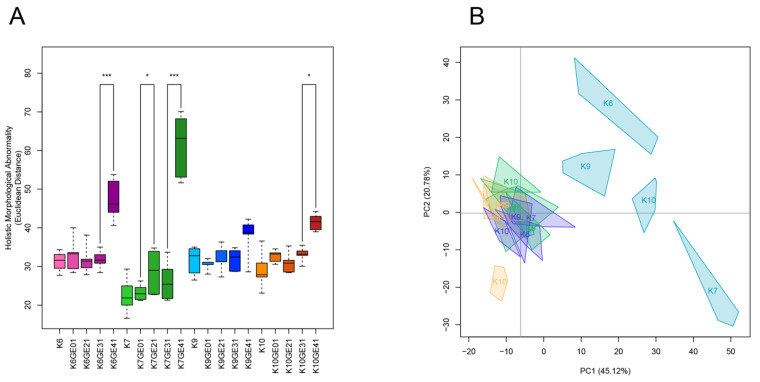
Morphological changes during genome-editing breeding. (**A**) Euclidean distances from their parental strain. The holistic morphological abnormality was calculated as the Euclidean distance of each replicate from the center of the parental strain in the orthogonal morphological space of the 44 PCs, capturing 90% of the variance of the 501 traits. For box and whisker plots, error bars are the standard deviation, the bottom and top of the box are the 25th and 75th percentiles, and the line inside the box is the 50th percentile (median). Asterisks (*, ***) indicate significant differences at FDR < 0.05, 0.001 (likelihood ratio test of one-way ANOVA with a gamma distribution), respectively. (**B**) Distribution of sake yeast isolates in the orthogonal morphological space. Principal component analysis was performed on 501 morphological traits of 16 genome-edited strains and their original sake yeast isolates. The colors of the strain names and regions are the same as in Figure 2A. The first two PCs (PC1 *x*-axis, PC2 *y*-axis) captured 65.9% of the variance, as shown in parentheses.

**Figure 6 microorganisms-11-01274-f006:**
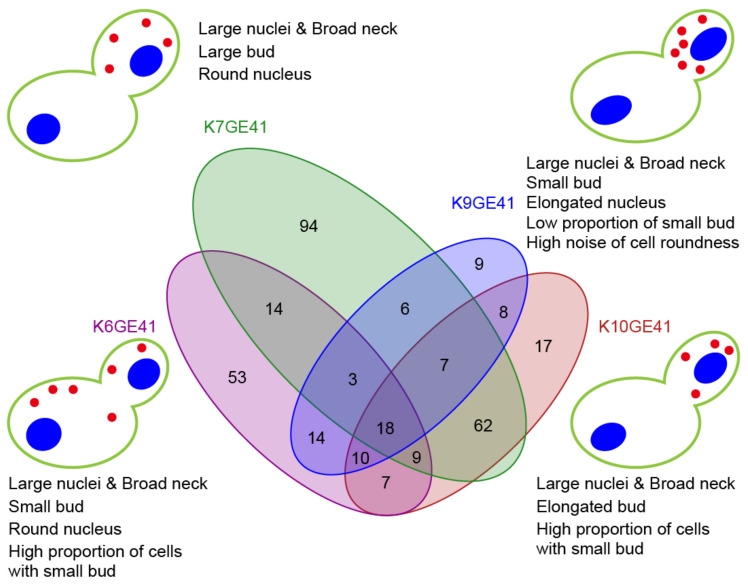
High-dimensional morphological changes at the step of the *FAS2*(G1250S)/*FAS2*(G1250S) introduction. A Venn diagram showing the number of the shared morphological traits changed. Wald tests were used to detect the parameters for which this step of breeding had a significant effect (FDR = 0.05). Among the 501 morphological parameters, 331 parameters were detected with this step. Schematic representation of the morphological features of the genome-edited sake yeast strains. One-way ANOVA of the generalized linear model was applied to 501 morphological traits of five sake yeast strains; 13 traits were detected at *p* < 0.05 after Bonferroni correction.

## Data Availability

Any additional data will be available upon request to the corresponding author.

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
