# Peer review of "Targeted Mutations Produce Divergent Characteristics in Pedigreed Sake Yeast Strains"

_microorganisms, 2023, doi:10.3390/microorganisms11051274_

Round 1
Reviewer 1 Report
The manuscript "Targeted mutations produce divergent characteristics in pedigreed sake yeast strains" by Klinkaewboonwong et al. examines the impact of genome editing of different sake yeasts on the characteristics of sake brewed with those yeasts. The paper provides valuable information despite the unintentional introduction of nonspecific mutations. To enhance the paper's impact, the following modifications are suggested:
1. Provide genotypes of each strain in a table for clarity (Lines 104-106).
2. Describe the composition of the media and culture methods (Line 106).
3. Briefly describe the methods for measurements (Lines 158-164).
4. Use proper formatting for numbers (Line 172).
5. Arrange the letters of the alphabet in Figure 2 in the order they appear in the diagram.
6. Revise the figure legend to explain the meanings of each mark (Lines 275-276).
7. Discuss the unexpected mutations introduced into the strains and if there was any genomic region near the sequence that showed similarity to the guide sequence used in the CRISPR-Cas system.
Author Response
We appreciate to the excellent reviewer for providing very helpful comments on our manuscript. We carefully read the reviewers’ comments one-by-one and made this revision.
- Provide genotypes of each strain in a table for clarity (Lines 104-106).
>We thank the reviewer for this comment. Accordingly, we added new supplementary Table (Table S1)
- Describe the composition of the media and culture methods (Line 106).
>We thank the reviewer for this comment. Accordingly, we described the composition of the media and culture methods in details (Line 121-126).
- Briefly describe the methods for measurements (Lines 158-164).
>We thank the reviewer for this comment. Accordingly, we described the methods for measurements (Line 194-203).
- Use proper formatting for numbers (Line 172).
>We corrected our careless mistakes (Line 212).
- Arrange the letters of the alphabet in Figure 2 in the order they appear in the diagram.
>We corrected our careless mistakes (new Fig. S6).
- Revise the figure legend to explain the meanings of each mark (Lines 275-276).
>We thank the reviewer for this comment. Accordingly, we corrected the legends of the Figure (new Fig. S6).
- Discuss the unexpected mutations introduced into the strains and if there was any genomic region near the sequence that showed similarity to the guide sequence used in the CRISPR-Cas system.
>Since there was no common denominator among the mutations introduced, we think that they were not due to off-targeting modifications by CRSPR/Cas9. We described this possibility in the text (Line 341-342).
Reviewer 2 Report
In this manuscript, targeted mutations were introduced into sake yeast strains, which exhibited divergent characteristics. In generally, the research features in this manuscript are not well-presented, and the main result does not catch the reader's eyes, sometimes the work is insufficient.
The quality of the manuscript needs to be further improved.
The problem was significant and concisely stated.
1) In the part of “2. Materials and Methods”, the expression of a series of “ we applied, we used, we extracted, we isolated...” is not appropriate, and the passive voice is recommended in scientific paper writing .
2) The presentation of Figure 1 (Breeding strategy of sake yeast) is relatively simple, the whole strategy of the research process can be plotted as Figure 1.
3) The very little information of Figure 2 presented for the readers, so I suggest, the Figure 2 could be replaced by the Table S7.
4) The 1-3 significant results or detailed improvements should be mentioned in the part of “Abstract”
5) The Figure 3 & 4 is so colorful that disturb the comfortable understanding of the detected results , should be changed. One or two table may be suitable for the presentation.
6) The Figure 5 “Principal component analysis (PCA) of sake produced by genome-edited yeast strains”, if it can be considered to move to the section of supplemental materials?
7) The part of "Conclusions", should be rewritten.
should be improved
Author Response
We appreciate to the excellent reviewer for providing very helpful comments on our manuscript. We carefully read the reviewers’ comments one-by-one and made this revision.1) In the part of “2. Materials and Methods”, the expression of a series of “ we applied, we used, we extracted, we isolated...” is not appropriate, and the passive voice is recommended in scientific paper writing .
>We thank the reviewer for this comment. Accordingly, we made all sentences  in “2. Materials and Methods” passive (Line 133,135,137,144,175,176,180,194-203,212,241,246,260, and 276).
2) The presentation of Figure 1 (Breeding strategy of sake yeast) is relatively simple, the whole strategy of the research process can be plotted as Figure 1.
>Relatively simple Figure 1 was presented to help readers to understand difference between the two previously mentioned approaches (the use of pedigreed sake yeast strains with different genetic backgrounds and the evidence-based approach). We added strain information in Figure legend. We also changed the text referring to Fig. 1B for greater accuracy (Line 84, 112, and 114).
3) The very little information of Figure 2 presented for the readers, so I suggest, the Figure 2 could be replaced by the Table S7.
>According to the reviewer’s suggestion, we moved Figure 2 to supplemental materials (Fig. S6). Table S7 was still included in the supplemental materials, because it is not the data strongly related to the subject of the paper.
4) The 1-3 significant results or detailed improvements should be mentioned in the part of “Abstract”
>We thank the reviewer for this comment. Detailed improvements in this study were to use several pedigreed sake yeast strains such as Kyokai strains No. 6, No. 9, and No. 10. Significant results were amino acid content and isobutanol differed among the strain backgrounds. We mentioned them in the part of “Abstract”.
5) The Figure 3 & 4 is so colorful that disturb the comfortable understanding of the detected results , should be changed. One or two table may be suitable for the presentation.
>We thank the reviewer for this comment. We agree with the reviewer that the colors in Figures 3 and 4 were too harsh. In the revised manuscript, we used mild colors, using color code as in Figure 1A like hot pink (K6), lime green (K7), deep sky blue (K9), and dark orange (K10). We still believe that simple bar graphs are easier to understand than a table with many numbers.
6) The Figure 5 “Principal component analysis (PCA) of sake produced by genome-edited yeast strains”, if it can be considered to move to the section of supplemental materials?
>We thank the reviewer for this comment. Accordingly, we move Figure 5 to the section of supplemental materials.
7) The part of "Conclusions", should be rewritten.
>We thank the reviewer for this comment. We added more information in the “Conclusion”.
Reviewer 3 Report
The following and other issues need to be addressed accordingly before this paper is considered for publication.
(1) Introduction: line 59-65 need citation.
(2) Please avoid results of the current study from the introduction section (see line 87-91).
(3) Please use uniformly the active or passive voice in the materials and methods section (could be applied in all sections of the manuscript) unless and otherwise it is mandatory.
(4) Italicize all the scientific names including Aspergillus oryzae (line 150).
(5) Authors need to mention in section 2.6 how the morphological data was obtained.
(6) It is not clear why the authors include this: “This section may be divided by subheadings. It should provide a concise and precise description of the experimental results, their interpretation, as well as the experimental conclusions that can be drawn.” (line 317-319)
(7) Figure 6: authors should indicate how these images were generated.
(8) Authors may include limitations of this study, future work, and recommendations in the conclusions section.
(9) Use a similar style to abbreviate the journal names in the references section.
Minor editing of English language required.
Author Response
We appreciate to the excellent reviewer for providing very helpful comments on our manuscript. We carefully read the reviewers’ comments one-by-one and made this revision.
(1) Introduction: line 59-65 need citation.
>We thank the reviewer for this comment. We added the following three papers cited in the introduction.
[9] Yoshida, M. Kyoukai Sake Yeast 13. J. Brew. Soc. Japan 1985, 80(9), 601-602, doi.org/10.6013/jbrewsocjapan1915.80.601
[10] Yoshida K. Kyoukai Sake Yeast 1801. J. Brew. Soc. Japan 2006, 101(12), 910-922, doi.org/10.6013/jbrewsocjapan1988.101.910
[11] Tamura, H., Okada, H., Kume, K., Koyano, T., Goshima, T., et al. 2015. Isolation of a spontaneous cerulenin-resistant sake yeast with both high ethyl caproate-producing ability and normal checkpoint integrity. Biosci. Biotechnol. Biochem.,79(7), 1191-1199.
(2) Please avoid results of the current study from the introduction section (see line 87-91).
>We thank the reviewer for this comment. We delete this part in “Introduction”.
(3) Please use uniformly the active or passive voice in the materials and methods section (could be applied in all sections of the manuscript) unless and otherwise it is mandatory.
>We thank the reviewer for this comment. Accordingly, we made all sentences  in “2. Materials and Methods” passive (Line 133,135,137,144,175,176,180,194-203,212,241,246,260, and 276).
(4) Italicize all the scientific names including Aspergillus oryzae (line 150).
>We fixed our careless mistakes(Line 184)..
(5) Authors need to mention in section 2.6 how the morphological data was obtained.
>We had mentioned how the morphological data was obtained in section 2.5. To make the link between section 2.5 and 2.6, We add a sentence in the text (Line 247).
(6) It is not clear why the authors include this: “This section may be divided by subheadings. It should provide a concise and precise description of the experimental results, their interpretation, as well as the experimental conclusions that can be drawn.” (line 317-319)
>We fixed our careless mistakes and delete the sentences.
(7) Figure 6: authors should indicate how these images were generated.
>We thank the reviewer for this comment. Accordingly, we added how these images were generated in the Figure legend (new Fig. 4, Line 532-534).
(8) Authors may include limitations of this study, future work, and recommendations in the conclusions section.
>We thank the reviewer for this comment. Accordingly, we added recommendation at the end of conclusion (Line 641-643).
(9) Use a similar style to abbreviate the journal names in the references section.
>We thank the reviewer for this comment. We fixed our careless mistakes (Line 739,741).